# The impact of long-term care needs on the socio-economic deprivation of older people and their families: A scoping review protocol

**Rossella Martarelli[1], Georgia Casanova[1,2]*, Giovanni Lamura[1]**

**1** Centre for Socio-Economic Research on Ageing, IRCCS-INRCA National Institute of Health & Science on Ageing, Ancona, Italy, **2** Instituto de Investigación en Políticas de Bienestar Social (POLIBIENESTAR)—Research Institute on Social Welfare Policy, Universitat de València, Valencia, Spain

* g.casanova@inrca.it

**Data Availability Statement:** No datasets were generated or analysed during the current study. All relevant data from this study will be made available

## Abstract

Major global problems such as population ageing, long-term care and the socio-economic burden of chronically ill older people and their families are urgent issues. Research in this field contributes to the growing international literature on health-related quality-of-life instruments, but little is known about the links between the related variables. Thus, the scoping review this protocol refers to plans to examine the socio-economic consequences of older people's poor health on their economic conditions and their families. In particular, the main aims are: a) to map the main concepts that characterize the body of the reference literature; b) to identify conceptual gaps or unexplored research areas to be addressed; c) to illuminate the difficulties that affect a large number of families with older members to care for, with particular attention to the concept of socio-economic deprivation, which includes material living conditions as well as social aspects (e.g. in the form of loneliness experienced as a consequence of health disorders). This protocol paper fulfils the purpose of clarifying the planned methodological phases, including the sub-phases, and listing the techniques used. A three-step approach is being applied, consisting of: pre-planning phase, protocol phase, and conduction and reporting phase. The preliminary stages of the protocol design are part of a dedicated project within the Open Science Framework platform and included in a Research Square preprint. This proposed project will contribute to multidisciplinary research on the connections between ill health and poverty, and could support critical reflections on the current evidence and guide future policies to alleviate this double burden.

## Introduction

The global population is getting older. In 2020, people aged at least 65 represented 9.3% of the worldwide population, and they are expected to increase to 16.0% by 2050 [1]. The oldest segment of the population, in particular, is on the rise: the so-called oldest-old (those over 80 years old) account for 6.3% of the total population in Europe, and their incidence is expected to double by 2050 [2, 3]. The number of older people in need of long-term care (LTC) is set to grow radically worldwide, together with the need for formal and informal care [4].

upon study completion. However, in accordance with the express purpose of enhance openness, transparency and reproducibility in science, we chose to facilitate the sharing of all our "procedural data" — i.e. all those produced so far — by creating a dedicated project within the Open Science Framework (OSF) platform. The project is now connected to the OSF Registries, a scholarly repository built for sharing data and searching registrations of research. Thus, this study is now registered as a project protocol and all its files – both figures and tables – are publicly accessible for broad dissemination (URL: https://osf.io/registries; DOI: 10.17605/OSF.IO/XQ58Z).

**Funding:** The study is being supported by the Family International Monitor (FIM) and the International Centre for Family Studies (CISF). Also, GC was supported by the the Marie Curie European Fellowship Grant (Grant Agreement No. 888102). Details: The scoping review this protocol refers to is part of the Horizon 2020-funded SEreDIPE project, i.e. the European Commission is to be considered as a funding body (call for proposal: H2020-MSCA-IF-2019; project code: 888102; project period: 16 March 2021 - 15 March 2023). SEreDIPE is an acronym for "Socio-Economic (SE) deprivation related to the effect of the presence of Dependent older people: strategies for Innovative Policies in Europe". Its mission is to investigate social trends and social innovation policies, with particular attention to informal care and the amount that families pay directly for looking after their older members, particularly those afflicted with limitations in activities of daily living (ADLs). POLIBIENESTAR, a research institute of the University of Valencia, is responsible for coordinating this project. The Italian National Institute of Health & Science on Ageing — IRCCS INRCA — is to be referred to as a consortium partner, the others being: Golgi Cenci Foundation (IT), University of Dortmund, Leibniz Institute of Social Sciences (DE), University of Vechta, CoE AgeCare (FI), "National Cancer Institute" Foundation (IT), European Center for Social Welfare Policy and Research, and Eurocarers (Belgium). Grant recipient: Dr. Georgia Casanova (Marie Curie Individual Fellowship Grant; H2020-MSCA-IF-2019. Grant Agreement: 888102). The study also partially benefits from funding for current research granted by the Italian Ministry of Health to the National Institute of Health & Science on Ageing (IRCCS — INRCA). https: www.inrca.it/ INRCA/MODM2/ FIM: https://www.facebook.com/ FamilyInternationalMonitor/ CISF: https://www.cisf. famigliacristiana.it European Commission: https:// cordis.europa.eu/project/id/888102/it The funders had and will not have a role in study design, data

Caring for the state of health of older people involves interventions at the biomedical and psychosocial levels and measures of economic interest [5], and all of them weigh on the balance of families. In particular, long-term care has been included among the highest priorities of national and international policy strategies on wealfare and health [6]. Many international research projects (e.g. EUROFAMCARE, ANCIEN, INTERLINKS, MOPACT, and Cequa) focussed their attention on LTC and its different aspects and key stakeholders. Considering their role as primary caregivers, families are essential stakeholders in the LTC context [7–9]. In 2017, Mitra et al. [10, 11] pointed out that the cost of LTC was high compared with incomes typical of many countries. Even in countries with advanced social protection systems, family-givers' care-related out-of-pocket expenses affect families' financial availability, even because more and more families purchase private care services. In addition to direct costs, they are involved in difficulties derived from a variety of indirect costs: the more time they spend caring for older people, the less they work, and this quickly results in a reduction in earning capacity [12–14].

There is no shortage of studies on the interplay between the socioeconomic deprivation of older people and their health conditions. Material deprivation reduces the possibility of coping with health needs and problems [15, 16] and influences both the psychosocial well-being [17] and the cognitive conditions of these people [18]. Moreover, as indicated in many studies, the combined effect of economic impoverishment and socio-relational deprivation creates remarkable differences in life expectancy, since wealthy seniors generally live longer [19, 20].

Nevertheless, it should be stressed that studies carried out in this field, while remarkable for the great attention paid to the effects of economic hardship on health conditions, often overlook the impact of health conditions on the socio-economic status (SES) of impaired or chronically ill people. Therefore, further studies on how health problems and related expenses affect the economic situation of families are necessary. Especially, the relationship between the LTC meaning, with this term referring to both formal and informal care, and the socio-economic deprivation of dependent older people and their families must be investigated. Promoting the interest of future research for this topic could facilitate an in-depth debate by the literature as well as an innovative approach to both social and economic policies. Guided by our intention to support research in this field and to contribute to its dissemination, we decided to conduct a scoping review (ScR) that mainly aims to: 1) systematically scan and evaluate the literature on the issue of dependent older adults with LTC needs and their SES; 2) identify any conceptual gaps as well as the most debated unsolved issues; 3) explore whether and how the "multidimensional perspective" on the concept of socio-economic deprivation (SED), as defined by Belletti & Casanova [21]; 4) identify the main outcomes achieved; 5) highlight the most useful insights for the policies to be applied as well as any suggestion for future research.

This type of review and the method of application allow us to meet these objectives and to provide a contribution to the literature on this topic. Moreover, this study will enable us to capture the meaning of the key concepts and definitions used in this field by researchers worldwide. In particular, we will be able to test the use of the multidimensional perspective in the concept of SED. This protocol paper aims to describe the methodological steps followed in our literature review process and help stimulate the production of new studies—both systematic and non-systematic reviews—on multidisciplinary research on such complex phenomena.

This study is part of a cross-national research effort promoted by the Family International Monitor (FIM). The research plan of FIM [22] aims to investigate how a condition of deprivation—understood as material and social deprivation—can affect families worldwide. The study is also part of the EU-funded SEreDIPE project (Horizon 2020 MSCA-IF-2019 Grant Agreement No. 888102), focused on the economic impoverishment of families with impaired older

collection and analysis, decision to publish, or preparation of the manuscript.

**Competing interests:** The authors have declared that no competing interests exist.

people and managed by the University of Valencia. Both projects deal with the concepts of "family" and "deprivation" from a multidimensional perspective (Belletti & Casanova; 2020).

## Materials and methods

This study refers to a composite but coherent set of methodological indications describing the rationale for undertaking a scoping review and the necessary procedures. In particular, the following references compose our methodological framework: a) the Lockwood et al. guidelines [23] on how to set up a ScR; b) the Munn et al. recommendations for structuring this type of study [24]; c) the Preferred Reporting Items for Systematic reviews and Meta-Analyses extension for Scoping Reviews [25]; d) the Joanna Briggs Institute (JBI) checklist [26]. The above-mentioned guidelines enable us to realize: a) all of the steps to be taken, i.e. 1) pre-planning; 2) protocol; 3) conducting and reporting; b) how to distinguish a ScR from a traditional or systematic review, i.e. how to achieve all of the objectives and present the results; c) how to sequence all of the study's stages and sub-stages; d) the type of information to be provided, such as the types of sources or 'effect measures' extracted from the selected studies (e.g. prevalence ratios for functional limitations; income inequality ratios; at-risk-of-poverty rate etc.). Finally, this protocol aligns with the Preferred Reporting Items for Systematic Reviews and Meta-Analyses for Protocols 2015 (S1 Checklist). S2 Fig shows a detailed representation of the steps and sub-steps of this ScR.

S1 Fig shows the conceptual framework used to realise the ScR and to identify the involved variables. The increasing incidence rate of common diseases and their subconditions constitutes a matter of public concern in both developed and developing countries, and represents one of the consequences of the population ageing phenomenon, and particularly of the extension of the oldest-old population segment. Older people's extended care needs contribute to a sharp increase in demands for informal care, the socio-economic burden of which is often borne by a family member. Macro-level demographic factors and institutional responses radically affect family dynamics at the micro-level, including the organizational adaptations and family adjustments covered by this ScR. Risk factors for SED among older people afflicted with ADL limitations and their families derive from two independent variables: a) the availability and affordability of LTC services for chronically ill older people; b) the probability of having to rely on informal caregiving and experiencing SED precisely because of the amount of family caregiving hours, mainly associated with chronic diseases whose medical care costs are hard to deal with.

### Preplanning

This stage involves formulating a series of broad questions: 1) What are the socio-economic conditions of families with impaired older people? 2) Is it true that the care of older dependent adults creates or intensifies financial burdens for their families? 3) How can family caregivers cope with the economic burden of care for the old? In response to these initial questions, we agreed that the ScR could help us to: 1) identify and draw up the boundaries of the main concepts used in literature, while also verifying their corrispondence to our conceptual framework and looking for any gaps in the reference literature; 2) investigate whether these families are so socio-economically deprived that they experience both a material economic impoverishment and social marginalization, or have some social support to mitigate such pressing difficulties; 3) identify the types of evidence and tools for measuring the healthcare-seeking behaviour of older people in need and their health requirements.

**Brainstorming.** The brainstorming phase allowed us to: a) identify in the ScR the most suitable type of study for answering our research questions; b) adjust and improve all the

points enumerated above. To begin with, each author assessed the structuring level of our initial questions and goals independently of the others, based on the several available scoping reviews and his/her own previous knowledge; this was followed by a brainstorming session which helped us to discuss our opinions, especially in relation to the possibility of addressing systematic reviews. We were in agreement on the main lines, i.e. that such a set of non-stringent questions and goals could be appropriate for an extensive investigation of the reference literature. Moreover, when compared to the seven goals listed by Lockwood et al. (S2 Fig), our initial purposes turned out to be very appropriate for a scoping review. We were able to confirm that most of those seven goals could be pursued and that this type of study could be undertaken. The brainstorming phase, i.e. an in-depth debate about the extent to which our questions should be defined, also allowed us to refine these and generate a final list of questions, which are: 1) to what extent are such families impacted by socioeconomic deprivation? 2) how much does the cost of care for the old contribute to the economic hardship faced by care recipients or families? 3) how thoroughly does the reference literature deal with these issues? 4) what are the most commonly found definitions of the "socioeconomic deprivation" concept in the reference literature? Finally, our initial purposes were slightly revised and definitively fixed on the basis of these 'new' questions.

**Definitive aims.** This study aims to: 1) identify the main key concepts and definitions used in the literature focused on the concepts described in S1 Fig; 2) pinpoint any little explored conceptual areas or countries where research needs to be strengthened; 3) examine how socioeconomic deprivation and related phenomena are conceptualized and debate whether the main definitions are consistent with our concept of "multidimensional deprivation"; 4) identify and report on the main existing results and insights provided, particularly those that stakeholders and policymakers can benefit from, e.g. in terms of social innovation.

This broad scope of objectives allowed us to filter through as many studies and evidence as possible, regardless of the countries in which the studies were conducted.

**Initial keywords.** A list of keywords has also been identified in the planning stage, which included: Long-term care, older (elderly) people, caregiver, family caregiving, impoverishment, deprivation, socioeconomic deprivation, economic, economic impact, poverty, and multidimensional poverty, since many researchers frequently use the term "poverty" as a synonym for "socioeconomic deprivation" in their articles. As detailed in the chapter referring to the "Protocol", these keywords were used to begin investigating all the relevant studies. The set of keywords is fully identified by the research team.

## Protocol

The protocol phase consists of: 1) a detailed explanation of the study phases, precisely the purpose of drafting this document; 2) a detailed explanation of the selection process: which studies and related research articles are concretely sought? What criteria do we refer to during the selection process? Since this is a scoping review, choosing broad criteria is strictly necessary (S2 Fig); therefore, we are only required to define three criteria: a) participants; b) concepts; c) context; 3) a detailed list of any information related to the study's development.

This document is the final version of our study protocol and results from revising two preliminary versions submitted to complete the OSF project (Open Science Framework) and Research Square webpages. Both can be accessed via:

- https://osf.io/xq58z. Registration DOI: 10.17605/OSF.IO/XQ58Z;

- https://www.researchsquare.com/article/rs-816117/v1.

  Resrach Square preprint: https://www.researchsquare.com/article/rs-816117/v1.

**Writing the protocol.**    This protocol is being drafted in accordance with the aforementioned guidelines. It mainly aims to illustrate the study's methodological framework and the types of materials and data to be used or obtained, such as: variables to be analyzed, gaps to be identified, and summary charts to be created. To this end, every effort has been made to make the study's content clear, methodologically reliable and adequately planned. The protocol also clarifies the link between questions and objectives (see previous paragraphs). Finally, it aims to describe how the unfinished steps will be completed.

**Selection process and eligibility criteria.**    Thus far, all articles were imported from the following databases (sorted by number of items found): PubMed, Wiley Online Library, Web of Science, and Scopus. We also consulted the main research library of the University of Cambridge (Cambridge University Library), which provided some useful references. All data is stored in cloud-based archives (Mendeley/EndNote X9).

The participants, concepts, context (PCC), and eligibility criteria that underpin this study are described below.

*Participants*. older adults in need of LTC and their family caregivers (both the family members who live with the older people and the ones who provide indirect assistance are included). We refer, in particular, to the so-called oldest-old people who cannot perform routine activities of daily living or self-care, regardless of their type of health problems. Moreover, the number of participants is not limited by the type of family or the country in which they live; both single-parent families (e.g. older people living alone) and married couples, with or without adult children, are included.

*Concepts*. a) the socioeconomic deprivation of impaired older people provided with LTC; b) the socioeconomic deprivation of their families. In detail: 1) "Older adults in need of (or provided with) LTC" are the ones affected by a reduced ability to perform routine activities, such as washing or bathing, dressing, feeding, transferring, and mobility, owing to age-related functional decline or chronic health problems. The fact that they depend on the treatment they need makes LTC indispensable; 2) LTC refers to services that help people in need of care, especially the old, with their physical and emotional needs over an extended period of time. LTC is by far the greatest health-care need of older adults and includes informal care [27]; 3) "informal care" is generally referred to as "unpaid care", mainly provided by a person connected to the older adult being taken care of, e.g. a spouse, child, or other relative. Nevertheless, the growing need for care among home-dwelling older adults is leading to the involvement of a higher number of people to support the old; 4) the term "family caregiving" represents the situation in which older adults are cared for by one or more family members, defining families in the broad sense, i.e. as individuals with a specific personal (S1 Fig). Family caregivers, in fact, can manage and provide home-based LTC both directly and indirectly; 5) the term "socioeconomic conditions" synthesizes a host of detectable factors, i.e. family assets, income, in work-benefits (if any), and savings and social ties related to the socio-demographic characteristics of family members. All of these are elements to which we must attribute the worsening or containment of LTC's economic burden. In the case of illness, the combined effect of a shortage or absence of these factors may alternatively lead to: a) "catastrophic health expenditure", which occurs when out-of-pocket health expenditure exceeds a certain ratio of household income to health spending; b) "economic impoverishment", which occurs when average household consumption falls below the international or national poverty threshold as a result of care expenses.

*Context*. all types of long-term care services for older people, especially informal care, in all geographical areas (both western and non-western countries). Nevertheless, the search process is conditioned by the considerable number of relevant studies conducted in low- and middle-income countries (respectively: LICs and MICs).

*Eligibility criteria*. in addition to the basic criteria illustrated above (PCC), it must be specified that: a) all studies investigating factors and policies that link activities, situations, or conditions such as old age, poor health, long-term care, health-related behaviours, and socioeconomic deprivation of families with older members affected by ADL limitations are included; b) studies proposing solutions to the economic problems triggered by health needs are particularly taken into account; c) special consideration is given to the ones outlining socially innovative solutions; d) all types of quantitative studies are included; e) qualitative studies and mixed methodologies, although scarce, are also included; f) all types of countries are considered in order to see whether there are any methodological differences that are somehow linked to the specific type of context; g) primary and secondary studies are considered; among the latter, systematic and scoping reviews are also taken into account; h) all selected relevant articles are no older than five years; exceptions to this rule are those selected due to the pertinence of the sources, of up to a maximum of ten years; i) empirical publications in peer-reviewed journals are preferred; j) all selected articles are written in English. Focus on English literature allows to include the main scientific literature, according to the most common strategy for literature review studies. The relevance of the topics studied for not English-speaking areas and countries represents however an invitation to realise future studies including different languages.

**Scoping review and evidence source details.**  The features characterising the scoping review and the sources referred to can be summarised as follows:

a. Scoping review title: Ageing, Long-Term Care, Poverty, and Socioeconomic Deprivation of Families: results from a scoping review;

b. Types of evidence sources: scientific journals; mainly research articles;

c. Details of evidence sources: we will list a number of items such as authors, publication dates, journal titles, volumes, and materials (if any) attached to the selected articles. All this information will be published at the end of the selection process;

d. The year of publication of the most recent sources among those selected: 2020;

e. Countries: we will also compile a list of countries where the selected studies were carried out. All this information will be published at the end of the selection process;

f. Types of evidence: most of the findings that we typically expect result from quantitative studies, often reinforced by the application of indicators (e.g. of disability or socioeconomic deprivation). For the most part, these studies turn out to be conducted on the basis of secondary data, resulting from both cross-sectional and longitudinal studies. Qualitative studies are uncommon. However, the research team will try to integrate qualitative and quantitative findings included in the study, in order to allow a more in-depth understanding of the phenomenon, as ensured by qualitative studies. As for quantitative studies, measurements of the probability of experiencing genuine economic impoverishment and social exclusion (after adjusting for age and income), as well as the degree of association between socio-relational deprivation and health problems, are generally the most common outputs;

g. Concepts and context of selected studies: the ways in which it is possible to care for older people in need of LTC in all countries, especially in middle- or low-income countries (MICs; LICs). The types of caregiver activities and the types of health problems of assisted persons are not indicated (bar a few exceptions). These studies mainly delve into: 1) the different shortcomings of national social protection systems; 2) the economic consequences

that affect families; 3) the role played by social skills in order to avoid falling into material and social deprivation;

h.  <u>Participants in the selected studies</u>: 1) caregivers: spouse or adult children (especially the ones cohabiting with the persons who need to be cared for, but this is not always specified); 2) care recipients: old, old-old and oldest-old impaired people; 3) families: all types of families (including those with adult children living elsewhere, although this is not clearly specified).

i.  <u>Details and results to be extracted from evidence sources</u>:

1.  Measurement properties of Activities of Daily Living scales (ADLs) used by most national surveys [28, 29], e.g. need assistance, received assistance, duration, special equipment, and perceived level of difficulty; 2) Items and measurement properties of the Frailty Index used by some of the selected studies (56-items FI), e.g. disabilities, self-reported health conditions, hearing, eyesight, cognitive function, and depressive symptoms; 3) Measurement properties of the Index of Multiple Deprivation 2004, used to identify the most and the least deprived areas in the UK.

## Conducting and reporting

This phase includes all the actions carried out or to be carried out at the operational level, from the beginning to the end of the study. Therefore, it also refers to the outputs of our first exploratory investigation. This initial step allowed us to identify the keywords we needed to use to perform a second selection, both to make a selection of chosen articles and to select new ones. A total of over ten new keywords were identified. We selected four main thematic areas to refer to in order to search for the most useful keywords: 1) family; 2) older people to be cared for; 3) assistance; 4) socioeconomic deprivation. Most of these new keywords pertain to the fourth area, e.g. healthcare expenditure, spending, payments, economic impoverishment/costs, burden, socioeconomic status (SES), and social differences. With regard to the first and the third, we added "household" and "informal (home) care", respectively. The selection of keywords resulted from the authors' debate, without any specialized support coming from research librarians. In total, 24 different keyword combinations (Tables 1 and 2) were used to carry out the exploratory phases described here; while the first step is to be referred to combinations 1, 3, 5, 7, 9, and 11, the secondary research phase, as well as the recursive selection process, derives from combinations 2, 4, 6, 8, 10, 12, 13, 14, 15, 16, 17, 18, 19, 20, 21, 22, 23, and 24. Each of the combinations was entered into one of the above-mentioned databases (PubMed, Scopus, Web of Science, Wiley Online Library, and Cambridge University Library).

While referring mainly to the aforementioned databases, other digital materials were also found, i.e. a book available online (using the following keywords: "abstract"; "family caregiving"; "economic impact") and an article extracted from the "JSTOR" library ("Proceedings. Annual Conference on Taxation and Minutes of the Annual Meeting of the National Tax Association"; search filters (3): 2018–2021; "only Journals"; refined by: "Family caregiving"). Non digital materials, including books on poverty-related issues, were also examined (two items, i.e. a book and an article published in the journal "On Intimacies: cultures and practices in current societies" — Issue 3/2017: working carers).

**Initial selection outputs.**  Thirty-five research articles were initially selected using PubMed and Scopus as a starting point, followed by Web of Science. This first selection included studies concerning the relationship between material and social deprivation and self-perceived well-being, or the connection between an early withdrawal from working life and

**Table 1. Keyword searching in PubMed.**

| | |
|---|---|
| 1 | (((caregiver[Title]) AND (poverty[Abstract])) OR (socioeconomic deprivation[Abstract])) AND (older people [Abstract])[1] |
| 3 | (((("2019/01/01"[Date—Publication]: "2019/12/31"[Date—Publication])) AND (long term care[Title])) AND (socioeconomic deprivation[Abstract])) OR (poverty[Abstract])[2] |
| 6 | (((poverty) AND (older people)) AND (informal care[Title/Abstract])) OR (home care[Title/Abstract])[3] |
| 8 | ((family caregiver[Title/Abstract]) AND (socioeconomic deprivation[Title/Abstract])) OR (poverty[Title/Abstract])[4] |
| 10 | ((((home care[Title/Abstract]) OR (informal care[Title/Abstract])) AND (older people[Title/Abstract])) OR (elderly[Title/Abstract])) AND disability[Title/Abstract])[5] |
| 11 | ((("Age and ageing"[Journal]) AND (long term care[Title])) AND (socioeconomic deprivation[Title/Abstract])) OR (poverty[Title/Abstract])[6] |
| 16 | ((("Australasian journal on ageing"[Journal]) AND (intergenerational)) OR (ageing)) AND (costs)[7] |
| 17 | ((((poverty[Title/Abstract]) OR (multidimensional poverty[Title/Abstract])) AND (disability[Title/Abstract])) OR (functional limitations[Title/Abstract])) AND (low-income countries)[8] |
| 19 | ((((((disability[Title]) AND (poverty[Title/Abstract])) OR (deprivation[Title/Abstract])) OR (economic costs [Title/Abstract])) AND (older people[Title/Abstract])) OR (elderly[Title/Abstract])) AND (low income)[9] |
| 20 | ((((poverty[Title]) OR (multidimensional poverty[Title])) OR (deprivation[Title])) AND (age[Title/Abstract])) OR (ageing[Title/Abstract])[10] |
| 21 | (((((socioeconomic[Title]) OR (socio-economic[Title])) AND (health[Title/Abstract])) OR (health problems [Title/Abstract])) AND (care[Title/Abstract])) OR (ADL limitations[Title/Abstract])[11] |
| 22 | (((life expectancy[Title/Abstract]) AND (social differences[Title])) OR (elderly[Title])) AND (socioeconomic status[Title/Abstract])[12] |
| 23 | (((((poverty[Title]) OR (healthcare expenditure[Title])) AND (income[Title/Abstract])) OR (low-income countries[Title/Abstract])) AND (deprivation[Title/Abstract])) OR (payments[Title/Abstract])[13] |
| 24 | (((((family caregiver[Title]) OR (older[Title])) AND (burden[Title/Abstract])) OR (socioeconomic status[Title/Abstract])) AND (activities of daily living[Title/Abstract])) OR (functional limitations[Title/Abstract])[14] |

[1]Filters (2): Abstract; Journal; [2]Filters: (5): Abstract; Journal Article; English; MEDLINE; Aged: 65+ years; [3]Filters (6): Article; Last 5 years; English; MEDLINE; Aged 65+; 80 and over; [4]Filters (5): Article; Last 5 years; English; Aged: 65+; 80 and over; [5]Filters (7): Article; last 5 years; English; 80+; 45+; 45–64; 65+; [6]Filters (4): Journal Article; from 2019/1/1 to 2019/12/31; English; MEDLINE; [7]Filters(4): Journal Article; time spam: from 2016/1/1 to 2019/12/31; English; MEDLINE; [8]Filters (3): Article; 1/1/2017-31/12/19; English; [9]Filters (3): Articles; last 10 years; English; [10]Filters (5): Article; English; 01/01/2019-present; 65+ and 80+ years; [11]Filters (3): Article; English; last 10 years; [12]Filters (3): 45–64 years; aged: 65+ years; time span: from 01/01/19 to 31/12/2019; [13]Filters (3): from 01/01/2019 to 31/12/20; article; English; [14]Filters (3): from 01/01/2019 to 31/12/20; article; English.

family caregiving. The last articles offered us many insights into the economic situation of women in particular, and the level of decreased earning capacity of family caregivers in general. In any case, most of the selected articles deal with the issue of health inequalities among older people; others emphasise that it is easily possible to fall into material deprivation even in high-income countries (HICs), when faced with severe health problems. It also does not seem to us to be insignificant that some of the first articles we found indicated an urgent need to thoroughly revise the methods and techniques for measuring the impact of ageing on the economic situation of countries.

**Second selection.** The additional keywords mentioned above turned out to be highly suitable as selection tools, since this second step allowed us to find many other relevant studies.

Together with those selected in the previous step, 63 research articles could be counted so far. The articles we are now finding are more focused on family economic difficulties and on how families act to ward off any type of economic impoverishment. What is striking is the fact that many older people around the world give up care or reduce it significantly. The drawing of a flow chart is planned in order to represent the entire search process in our final article.

**Table 2. Keyword searching in the remaining databases.**

| | |
|---|---|
| 2 | "intergenerational" anywhere and "ageing" anywhere and "expenditure" anywhere published in the "Australasian Journal on Ageing[1] |
| 4 | keywords to enter: ageing; generational; spending; family[2] |
| 5 | (ABS(*expenditure*) AND KEY (*older* AND *people*) AND ABS (*family*))[3] |
| 7 | TOPIC: (impoverishment) *AND* TOPIC: (household) *AND* TOPIC: (caregiver) *OR* TOPIC: (deprivation) *OR* TOPIC: (poverty) *AND* TOPIC: (elderly) *AND* TOPIC: (aged) *AND* DOCUMENT TYPES: (Article) *AND* LANGUAGE (English)[4] |
| 9 | (TITLE-ABS-KEY ("older AND people" OR elderly) AND TITLE-ABS- KEY ("household AND impoverishment") OR ABS (deprivation) AND KEY (economic)) AND (LIMIT-TO (SUBJAREA,"MEDI") OR LIMIT TO (SUBJAREA,"SOCI"))[5] |
| 12 | ""informal care" OR "home care"" anywhere and ""older people" OR "elderly"" anywhere and "carers" in Abstract [6] |
| 13 | ""informal+care"+OR+"home+care"" anywhere and ""older people" OR "elderly"" anywhere and "carers" in Abstract [7] |
| 14 | "economic" anywhere and ""older people" OR "aged"" anywhere and "family" anywhere and "caregivers" in Abstract and "intergenerational" anywhere[8] |
| 15 | ""poverty"+OR+"multidimensional poverty"" in Abstract and "health" OR "informal care" OR "long term care" in Abstract[9] |
| 18 | ""informal care" OR "long term care"" in Abstract and "family" anywhere and ""aged" OR "elderly"" in Abstract and "carers" anywhere[10] |

[1]**Wiley Online Library**; [2]**Cambridge University Press**. Filters (3): Journal "Ageing & Society"; 2016–2021; "only show content I have access to"; [3]**Scopus**. Filters: not applied; [4]**Web of Science**. Categories: (health care sciences services OR sociology OR health policy services OR social issues). [5]**Scopus**; [6]**Wiley Online Library**. Filters (3): 2012–2021; *Health&Health care*; Journals; [7]**Wiley Online Library**. Filters (5): journals; all dates; Health Economics; Australasian Journal on Ageing; Scandinavian Journal of Caring Sciences; [8]**Wiley Online Library**; [9]**Wiley Online Library**. Filter: 2015–2010; [10]**Wiley Online Library**. Filter: 2015/2019.

This graph will show how many articles were definitively selected, as well as how many were rejected and why.

**Quality assessment: Data extraction.** Once the search process is finalised, we will begin to take note of the studies' main features with the intention of identifying the extent to which the studies explain their aims, methods, and results. This process, which is referred to as "quality assessment", consists of two sub-stages: 1) **data extraction**: data will initially be extracted as text variables using a dedicated data extraction instrument (S1 Table). All the main features of each article will be summarized on the basis of a classification grid made up of the following items: a) name of the authors; b) title; c) year of publication; d) keywords; e) type of study (e.g. national or government census, questionnaire-based survey); f) type of methodological framework (e.g. quantitative methods); g) type of techniques used in the data analysis phase (e.g. statistical analyses, type of interviews, focus groups); h) reporting of methods, i.e. how sampling methods and measurement tools are described; i) aims and type of results provided by the authors; j) type of country to refer to. Finally, all these articles will be labelled as "high relevance, medium relevance, or sufficiently relevant studies" (with respect to our purposes); 2) **data coding**: after being illustrated with a text string, each of the options of the aforementioned items will be summarized in a numeric code, or rather, paired off with a numerical label representative of a certain concept to refer to, e.g. "1" if the study is a questionnaire-based survey, "2" if the study is a secondary data-based survey, and so on (item 5). Item 6, which refers only to the methodological framework of the selected studies, will be represented like this: "1" if only quantitative methods are available, "2" if only qualitative methods are available, "3" if only mixed methods are available. This coding process transforms each item from a mere property to a variable. Such a process, which is usually referred to as "data coding", is the most systematic way of synthesizing and re-presenting information and empirical objects. Once all the extracted information is each assigned a code, it will be possible to carry out a statistical-descriptive analysis. Microsoft Excel 2007 will be used to enter all coded data into a searchable database.

**Data analysis.** We agreed to conduct a mixed method analysis, based on a descriptive quantitative analysis integrated by a qualitative (contents) analysis, with the intention of identifying and tracing the boundaries of overlapping or unexplored conceptual areas. We will also

**Table 3. Conducting and reporting: The third phase of the study and related outputs.**

| Sub-phase | | Description | Output/s |
|---|---|---|---|
| 3A | *INITIAL* SELECTION | Examination of a broad area of knowledge on the issue of family health care spending and socioeconomic deprivation (SED) | **35** items selected |
| | | | Identification of key concepts and keywords (over ten new useful keywords) |
| 3B | *SECOND* SELECTION | Refinement of the search process through the new keywords identified (**choosing selected items**) | At present: **63** items collected |
| | | Additional items to be found (**advanced search**) | |
| 3C | QUALITY ASSESSMENT | Data quality assessment (what kind of data/results?) | List and classification of the main features of selected articles (**data extraction**); *JBI* data extraction tool (text variables) |
| | | | Codification of all extracted information (**data coding**). Data storage: *Microsoft Excel* |
| | | | Identification of the type of analysis (what kind of variables can be analyzed?) |
| 3D | DATA ANALYSIS | **Bibliographic Documentary Analysis** (combining automatic and manual procedures within the analysis process) | Identification of any conceptual gap and/or redundancy in research or basic knowledge |
| | | Meta-analysis? | Identification of any significant differences due to the country in which the selected studies were carried out |
| | | | Identification of data relationship sets and numerical measures (in the case of meta-analysis) |
| 3E | DATA SYNTHESIS and CONCLUSIONS | All factors linked to the relationship between family caregiving and **material and social deprivation** will be listed | Summary tables |
| | | Representation of the level of **economic impoverishment** differentiated by country | Graphics |

perform basic statistical analyses, such as univariate or bivariate analyses, in order to get the frequency distribution of some of the aforementioned variables and to obtain cross tabs, e.g. referring to the comparison between HIC care systems and LIC care systems. We will use the latest version of Maxqda to analyze all qualitative data. We will also try to conduct a meta-analysis, if possible, in order to combine all the main results concerning LICs (e.g. the tendency for health inequalities to rise) with the results of other countries, especially with regard to the problem of economic impoverishment as a result of the state of health.

**Data synthesis and reporting.** To summarize the main findings, all data will be represented in table form and, when appropriate, in graphic form, in a manner that aligns with the purposes of this ScR. A narrative summary will describe how this study can be fully reproduced and how the data answers the research questions. Not too many images will be inserted, in accordance with Munn et al. The main significant differences between low-, high-, and middle-income countries, in terms of out-of-pocket medical expenses, life expectancy, and the working conditions of family caregivers, will be presented as a summary to help illustrate the different political contexts in which the problem of "unpaid care" can emerge (see "Table 3").

## Discussion

We mentioned the paucity of publications on the economic burden of care for older people requiring LTC and the impact of health expenditure on their families' economic situation. However, the term "paucity" does not imply that available studies on this issue are scant. Indeed, when compared to the multitude of studies on the poor state of health of deprived older people, they are not enough for a truly exhaustive investigation: there is a compact two-way relationship between health and socioeconomic deprivation, and both directions need to be studied in-depth. We would also like to draw the attention of stakeholders and policy-makers to the general issue of increasingly burdensome health costs and the difficulty of

quantifying the number of people whose economic impoverishment and socio-relational deprivation are due solely or primarily to health spending. Many people, including older adults, refuse or partially refuse treatment, i.e. they are incorrectly estimated to be living above the so-called 'poverty line'. Thus, a rough estimate of the worldwide incidence of poverty is being discussed. This issue contributes to the urgency of our study, but at the same time it should be considered as a limitation, since we cannot confirm the number of people who turn out to be less deprived than they appear to be, nor how many people would be officially classified as 'poor', i.e. economically impoverished, had they received or agreed to receive appropriate assistance. This paper shows the path to follow to carry out a thorough ScR on the socio-economic deprivation of older ill people and their families precisely because it serves as a flexible, although structured, search map. A brief overview of what has been done so far is enough to see how this issue is scarcely present in the reference literature, despite its relevance to the public debate. This research protocol supports knowledge-sharing practices and underlines the necessity of following a complete and integrated methodological cycle, including the preliminary stages. The long design processing time, a limit that needs to be made explicit, is due to the intricate relationships between the core issues examined and those related to them. On the other hand, any effort to promote future research necessarily implies precise methodological schemes. Therefore, we decided to adopt a multi-stage design, aligned with all of our methodological criteria.

## Supporting information

**S1 Checklist. PRISMA-P 2015 checklist.**
(DOCX)

**S1 Table. Data extraction tool.** A, Study details and characteristics; B, Details or results to be extracted.
(DOCX)

**S1 Fig. Research concepts and goals.**
(JPG)

**S2 Fig. The study's methodological framework.**
(DOC)

## Acknowledgments

The authors are grateful to Dr. Francesco Belletti, Director of the International Centre for Family Studies (CISF), for his suggestions on how to develop the conceptual framework of the study.

## Author Contributions

**Conceptualization:** Georgia Casanova.

**Data curation:** Rossella Martarelli, Georgia Casanova.

**Funding acquisition:** Georgia Casanova.

**Methodology:** Georgia Casanova.

**Validation:** Rossella Martarelli, Georgia Casanova, Giovanni Lamura.

**Writing – original draft:** Rossella Martarelli, Georgia Casanova.

**Writing – review & editing:** Giovanni Lamura.

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
