## [Decision Letter · Decision Letter 0]

24 Jun 2022

PONE-D-22-09758The impact of long-term care needs on the socio-economic deprivation of older people and their families: A scoping review protocolPLOS ONE

Dear Dr. Casanova,

Thank you for submitting your manuscript to PLOS ONE. After careful consideration, we feel that it has merit but does not fully meet PLOS ONE’s publication criteria as it currently stands. Therefore, we invite you to submit a revised version of the manuscript that addresses the points raised during the review process.

Most of the comments that the reviewers have made are easily fixable. Changing specific words questioned by the reviewers are optional.

We look forward to receiving your revised manuscript.

Kind regards,

Edward Jay Trapido, ScD

Academic Editor

PLOS ONE

Journal Requirements:

Reviewers' comments:

Reviewer's Responses to Questions

**Comments to the Author**

1. Does the manuscript provide a valid rationale for the proposed study, with clearly identified and justified research questions?

Reviewer #1: Yes

Reviewer #2: Yes

2. Is the protocol technically sound and planned in a manner that will lead to a meaningful outcome and allow testing the stated hypotheses?

Reviewer #1: Yes

Reviewer #2: Partly

3. Is the methodology feasible and described in sufficient detail to allow the work to be replicable?

Reviewer #1: Yes

Reviewer #2: No

4. Have the authors described where all data underlying the findings will be made available when the study is complete?

Reviewer #1: Yes

Reviewer #2: Yes

5. Is the manuscript presented in an intelligible fashion and written in standard English?

Reviewer #1: Yes

Reviewer #2: No

6. Review Comments to the Author

You may also provide optional suggestions and comments to authors that they might find helpful in planning their study.

Reviewer #1: Thank you for asking me to review this protocol for a scoping review. The review aims to answer an important questions and an appropriate method is described.

Was the search strategy developed in conjunction with a research librarian? It would be useful to know this or why they were not involved.

Reviewer #2: Thank you for the opportunity to review this paper. I want to commend the authors on the focus of their study. This is a very interesting area for a review and could add new insights into the barriers to sufficient care of older adults. In general, a stronger scientific approach and more focused content is needed. From the outset it is unclear why a scoping review was chosen over a systematic review. Language issues were highlighted in-text and I added suggestions that could support you in clarifying the contents of the paper and strengthening its academic value.

7. PLOS authors have the option to publish the peer review history of their article (what does this mean?). If published, this will include your full peer review and any attached files.

Reviewer #1: **Yes: **William Gibson

Reviewer #2: No

---

## [Author Response · Author response to Decision Letter 0]

6 Aug 2022

Responses to Reviewer 1: 

Comment 1: The authors are grateful for your interest and study. Many thanks for your question. 

Comment 2: The study was conducted by an Italian team in which no research librarians were involved, as within the Italian context this position is not very common. However, in the section on keywords, we have specified that the keywords used have been carefully identified by a research team. This lack of research librarian support has been declared in the text in the Conducting and Reporting section.

Responses to Reviewer 2 

Comment 1: The authors are grateful to the reviewer for his/her detailed revision.

Comment 2: All suggestions have been checked, evaluated and included in the text. 

Comment 3: Checked and accepted. 

Comment 4: The sentence has been changed to underline the relevance of LTC for welfare and health policy strategies. 

Comment 5: The term “actors” has been changed into “stakeholders”.

Comment 6: The sentence has been revised to underline the impact of out-of-pocket expenditure on the financial availability of these families. 

Comment 7: Changed word with “purchase”.

Comment 8: Thanks, the authors appreciate your agreement.

Comment 9: The sentence has been rewritten to improve the clarity of the concepts included and the readability of the text.

Comment 10: The sentence has been rewritten to improve the clarity of the concepts included and the readability of the text.

Comment 11: Suggestion accepted by the authors; the sentence has been changed. 

Comment 12: Suggestion accepted by the authors; the sentence has been changed.

Comment 13: A larger definition of multidimentional deprivation is included in the Methods section. However, the sentence has been revised and the main reference for the concept has been added.

Comment 14: The sentences have been rewritten to better underline that the scoping review study focused on the multidimensional concept of socio-economic (SE) deprivation.

Comment 15: The reference to the multidimensional concepts of “family” and “poverty” has been introduced in the aims of the study (line 2), as previously suggested by you (suggestion 13). 

Comment 16: We would like to introduce the main references of our methodological framework. The sentence has been changed to better clarify this aspect.

Comment 17: Yes, the sentence has been changed and only the acronym has been included in it. 

Comment 18: The sentence has been changed to valorize that Fig. 1 shows the conceptual framework of the scoping review, useful to identify the involved variables. 

Comment 19: The paragraph has been moved as suggested by you. 

Comment 20: We agree that it should be modified, many thanks for your advice. The question has been reformulated, following what you suggested. 

Comment 21: This sentence has been changed to underline the contribution of the ScR to the mapping of concepts and detectable omissions. 

Comment 22: Changed.

Comment 23: The sentence has been rephrased.

Comment 24: Changed as suggested by you.

Comment 25: The question has been rewritten to be more clearly addressed to older care recipients and their families. 

Comment 26: The sentence has been changed to better underline the connection with the concepts used in literature and included in the conceptual framework.

Comment 27: The sentence has been changed. 

Comment 28: The paragraph has been checked and changed using the present tense. 

Comment 29: The paragraph has been changed according to your suggestion.

Comment 30: The authors thank you for this advice and agree with you on the relevance of the local literature to the topic. The authors also agree that this problem represents one of the key issues for conducting a literature review. Unfortunately, within this project we are unable to conduct a broader search with a different language, because local and mother-tongue teams must be involved to carry it out, and the available budget and timetables do not allow it. For these reasons, the authors decided to adopt the most common strategy in literature review studies, and to focus on English literature only, as this is the most widely used scientific language. Considering also the relevance and the scarce diffusion of similar studies on these themes, the authors hope that this study could stimulate future studies including literature in other languages. These relevant suggestions have been used to integrate the text into the main file, aiming to explain the reasons of this limitative choice. 

Comment 31: The choice relating to the kind of review to be carried out has been debated in the preliminary phases of the study design, as detailed in the paragraphs “Pre-planning” and “Brainstorming”. According to what is proposed by the guidelines adopted (Lockwood et al. 2019; Munn et al. 2018), independently of using quantitative or qualitative data - if the research questions are open and not too specific - it is recommended that researchers carry out a ScR, rather than a systematic review (see the seven aims listed in Fig. 2). The need to better understand the concepts involved in the conceptual framework and to understand how literature uses them, led us to start our reflection with a ScR. However, conducting a scoping review does not exclude that a systematic review can be carried out on the same issue in the future. 

Comment 32: Many thanks for this advice. In this section we explain the different data sources. The useful contribution to a better understanding of phenomena coming from the integrated use of both quantitative and qualitative studies has been underlined by an additional sentence.

Comment 33: The word has been changed into “screened” as suggested by you.

Comment 34: We checked the text again. 

Comment 35: Many thanks for your advice. What we assessed here was not the quality of research but methods reporting, i.e. how the methods chosen have been described. The sentence in the text has been changed to better clarify this aim. 

Comment 36: The data analysis section has been revised and integrated to underline the mixed method analysis to be used in this study: descriptive quantitative analysis (based on frequency analysis) and contents qualitative analysis (to explore contents of results).

---

## [Editor Report · Decision Letter 1]

16 Aug 2022

The impact of long-term care needs on the socio-economic deprivation of older people and their families: A scoping review protocol

PONE-D-22-09758R1

Dear Dr. Casanova,

We’re pleased to inform you that your manuscript has been judged scientifically suitable for publication and will be formally accepted for publication once it meets all outstanding technical requirements.

Kind regards,

Edward Jay Trapido, ScD

Academic Editor

PLOS ONE
---

## [Editor Report · Acceptance letter]

22 Aug 2022

PONE-D-22-09758R1 

The impact of long-term care needs on the socio-economic deprivation of older people and their families: A scoping review protocol 

Dear Dr. Casanova:

I'm pleased to inform you that your manuscript has been deemed suitable for publication in PLOS ONE. Congratulations! Your manuscript is now with our production department. 

Kind regards, 

on behalf of

Dr. Edward Jay Trapido 

Academic Editor

PLOS ONE